

# Preoperative serum lipids as novel predictors for concomitant thyroid carcinoma in Graves' disease

Xingxing Gao[1,*], Mi Liu[2,*] and Yijun Wu[1]

[1] Department of Thyroid Surgery, The First Affiliated Hospital, Zhejiang University, Hangzhou, China
[2] Center for Rehabilitation Medicine, Department of Anesthesiology, Zhejiang Provincial People's Hospital, Affiliated People's Hospital, Hangzhou Medical College, Hangzhou, China
* These authors contributed equally to this work.

## ABSTRACT

**Background:** The occurrence of thyroid carcinoma in patients with Graves' disease (GD) has been rising recently. However, the linkage between lipids and the incidence of thyroid carcinoma among GD patients is still not well-established.
**Objective:** The research aims to explore the relationship between serum lipid concentrations and the occurrence of thyroid cancer in patients diagnosed with GD.
**Methods:** We conducted a retrospective analysis of data from 512 patients with GD who underwent surgical procedures at our institution between 2015 and 2024. Our study focused on examining the correlations between various patient characteristics and the occurrence of thyroid cancer. Logistic regression models were developed to analyze the predictive factors. Ultimately, we constructed a predictive nomogram to estimate the potential of thyroid cancer in GD patients.
**Results:** Among the 512 patients with GD, 299 patients were pathologically confirmed as differentiated thyroid carcinoma (DTC) (58.4%). Multivariate analysis revealed that high triglyceride (TAG > 1.185 mmol/L), low high-density lipoprotein (HDL < 1.325 mmol/L), and overweight (body mass index (BMI) ≥ 25) were risk factors for malignancy. In addition, ultrasound characteristics, including nodules in the thyroid, aspect ratio imbalance, hypoechogenicity, irregular borders, and microcalcifications, were risk factors for malignancy. The predictive nomogram demonstrated significant clinical utility, exhibiting an area under the curve (AUC) of 0.91 (95% CI [0.88–0.94]) and 0.91 (95% CI [0.87–0.96]) in the training set and validation set. Moreover, a high level of TAG was a risk factor for central lymph node metastasis and high AJCC staging in GD patients with thyroid carcinoma.
**Conclusions:** Our study presents initial findings suggesting that elevated TAG levels, reduced HDL cholesterol levels, and overweight status are individually linked to the incidence of thyroid carcinoma in patients with GD. These results indicate that preoperative serum lipid profiles and BMI can serve as valuable predictors for the occurrence of thyroid carcinoma in this patient population.

Corresponding author
Yijun Wu, wuwu5925@zju.edu.cn

## INTRODUCTION

Graves' disease (GD), an autoimmune disorder, stands as the most prevalent cause of hyperthyroidism in regions with adequate iodine intake (*Smith & Hegedüs, 2016*). This condition arises due to the formation of thyroid-stimulating hormone receptor antibodies (TRAb), which provoke the thyroid gland to secrete excessive thyroid hormones, resulting in hyperthyroidism (*McIver & Morris, 1998*). Despite antithyroid medications (thionamides) being the primary treatment option globally, a definitive therapeutic intervention, such as radioiodine (RAI) or thyroidectomy, is often necessary due to the high incidence of relapse following thionamide discontinuation or based on patient preference and other considerations (*Bartalena, Chiovato & Vitti, 2016*). Resent research reported that patients with Graves' disease have a high risk of thyroid malignancy (*Keskin et al., 2019*; *Pazaitou-Panayiotou, Michalakis & Paschke, 2012*). Differentiated thyroid carcinoma, arising from epithelial follicular cells of the thyroid, accounts for approximately 95% of all thyroid cancer cases and encompasses papillary thyroid carcinoma (PTC), follicular thyroid carcinoma (FTC), and Hurthle cell carcinoma (*Jung, Bychkov & Kakudo, 2022*). The correlation between GD and concomitant thyroid carcinoma remains contentious. Several studies have indicated a higher incidence of thyroid carcinoma in GD patients compared to the general population (*Wahl et al., 1982*; *Belfiore et al., 1990*; *Kraimps et al., 2000*). While the link between GD and thyroid carcinoma remains elusive, GD can complicate the diagnosis and management of thyroid nodules (*Durante et al., 2018*; *Belfiore et al., 2001*; *Arslan et al., 2000*). For GD patients with thyroid cancer, surgical planning typically follows the diagnosis of nodules *via* ultrasound or fine-needle aspiration biopsy (FNAB). Nevertheless, thyroid cancer is occasionally detected incidentally during postoperative pathological examination (*Phitayakorn & McHenry, 2008*; *Dănilă et al., 2008*; *Jia et al., 2018*). These instances underscore the necessity of surgery and suggest that cancer might have been overlooked had surgery not been performed for other indications. Therefore, the identification of biomarkers for concurrent thyroid carcinoma in GD patients could facilitate preoperative planning and enhance screening thoroughness, even in the absence of preoperative nodule detection.

Prior research has suggested a significant correlation between obesity and dyslipidemia with an elevated risk of thyroid carcinoma development. However, there is a scarcity of studies examining the relationship among obesity, blood lipid concentrations, and thyroid cancer risk specifically in GD patients. A study involving 122 GD patients revealed that those diagnosed with thyroid cancer exhibited a notably higher body mass index (BMI) compared to their counterparts without thyroid carcinoma (*Park et al., 2024*). This observation is particularly intriguing given that hyperthyroidism in GD patients typically results in weight reduction and lower blood lipid levels. Consequently, there is a pressing need for further investigation to delve deeper into the relationship between obesity, dyslipidemia, and the risk of thyroid cancer in the GD patient population.

In this study, we retrospectively recruited a cohort of 519 patients diagnosed with GD, along with their comprehensive clinicopathological data. We conducted an analysis to explore the preoperative lipid profiles in relation to the risk of differentiated thyroid carcinoma (DTC) among these GD patients. Additionally, we undertook further analyses

aimed at developing a prediction model based on serum lipids, integrated with clinical characteristics.

## MATERIALS AND METHODS

### Patients

We collected the retrospective data of 577 patients with GD who underwent thyroidectomy spanning from January 2015 to December 2024 at The First Affiliated Hospital, Zhejiang University School of Medicine (depicted in Fig. 1). Studies involving human participants were reviewed and approved by the Clinical Research Ethics Committee of the First Affiliated Hospital College of Medicine, Zhejiang University (Approval number: 2024-1193). The data are anonymous, and the requirement for informed consent was therefore waived. The diagnostic criteria for GD encompassed biochemical evidence of hyperthyroidism, the presence of serum anti-thyrotropin receptor antibodies, increased diffuse uptake of $^{123}$I or $^{99}$mTc-pertechnetate on radionuclide scans, and/or the manifestation of clinical features such as goiter or Graves' ophthalmopathy, as outlined in Ross et al. (2016). Exclusion criteria for this retrospective study encompassed: (1) patients diagnosed with anaplastic or medullary thyroid cancer; (2) those lacking comprehensive clinical data; (3) individuals with a prior history of thyroid surgical intervention; (4) patients without complete laboratory testing for serum lipid and thyroid hormone levels; and (5) patients harboring malignancies in other organ systems. The study protocol was granted approval by the Ethics Review Board of the First Affiliated Hospital, Zhejiang University School of Medicine.

Patients' clinical information, pathological characteristics, and laboratory results were extracted from their medical records. The clinical information comprised patients' age, gender, duration of GD, BMI, comorbidities like hypertension and diabetes mellitus, and ultrasound features of thyroid nodules, including aspect ratio (anteroposterior to transverse diameter), margin definition, shape, echogenicity, and calcification presence. Pathological data encompassed tumor size, multifocality, bilaterality, extrathyroidal extension (ETE), lymph node metastases (LNM), and staging according to the 8th edition of the International Union Against Cancer (UICC)/American Joint Committee on Cancer (AJCC) tumor node metastasis (TNM) classification. Laboratory data encompassed preoperative serum lipid levels (triglycerides (TAG), total cholesterol (TC), high density lipoprotein (HDL), low density lipoprotein (LDL), very low density lipoprotein (VLDL)) and thyroid function indicators such as thyrotropin, antithyroglobulin antibodies (TgAb), thyroid peroxidase antibody (TPOAb), thyroglobulin (Tg), and thyroid-stimulating hormone receptor antibody (TRAb). Overweight status was defined as a BMI of 25 kg/m$^2$ or greater.

### Statistical analysis

Continuous variables were summarized as mean value with standard deviations (SDs), whereas categorical variables were presented as counts and percentages. Categorical variables were compared using the chi-square test. Receiver operating characteristic (ROC) curves were constructed, and optimal cutoff values for serum TAG, TC, HDL, LDL, and

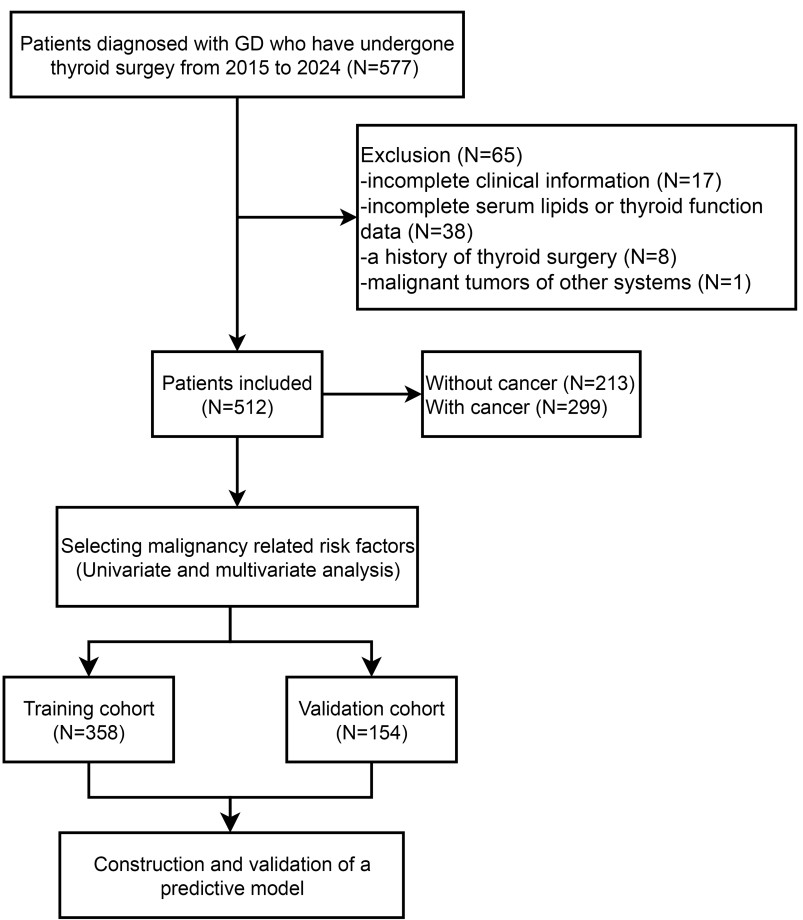

**Figure 1** Flow diagram of patient selection and data analysis.

VLDL were determined using the Youden index. The cut-off value for TAG, TC, HDL, LDL, and VLDL was 1.185, 3.805, 1.325, 2.305, 0.515 mmol/L, respectively. Logistic regression analysis was employed to identify risk factors for thyroid carcinoma in GD patients. A nomogram was developed based on the logistic regression model to predict DTC in GD patients. Calibration plots were used to compare the actual diagnoses with the predicted probabilities. ROC curves were plotted, and area under the curve (AUC) values were computed to assess the predictive performance of the nomogram model. All statistical analyses and interaction effect evaluations in this study were conducted rigorously in accordance with statistical methodologies. Statistical computations were executed using SPSS version 26.0 (IBM Corp., Armonk, NY, USA). The nomogram was created in R software (R Foundation, version 4.0.3) utilizing the "rms" package.

# RESULTS

## Baseline clinicopathological characteristics of patients with Graves' disease

The baseline characteristics of patients enrolled in the study were summarized in Table 1. Among the 512 patients with GD, 299 patients were pathologically confirmed as DTC

**Table 1 Baseline clinicolpathological characteristics of the study population.**

| Variables | Total ($n$ = 512) |
|---|---|
| Age | 45.22 ± 13.62 |
| Gender | |
| Male | 84 (16.22) |
| Female | 428 (83.59) |
| Smoking | |
| Yes | 13 (2.54) |
| No | 499 (97.46) |
| Drinking | |
| Yes | 6 (1.17) |
| No | 506 (98.83) |
| Hypertension | |
| Yes | 75 (14.65) |
| No | 437 (85.35) |
| Diabetes | |
| Yes | 17 (3.32) |
| No | 495 (96.68) |
| Duration of GD | 55.35 ± 79.19 |
| Preoperative management | |
| KI | 23 (4.49) |
| MMI | 454 (88.67) |
| PTU | 15 (2.93) |
| None | 20 (3.91) |
| Cause of surgery | |
| Refractory, complication to medication | 91 (17.77) |
| Huge goiter | 154 (30.08) |
| Ophthalmopathy | 26 (5.08) |
| Preoperative dx of cancer or thyroid nodule | 241 (47.07) |
| Extent of surgery | |
| Lobectomy | 48 (9.38) |
| TT/near TT | 464 (90.62) |
| BMI | 23.01 ± 3.39 |
| Serum lipids | |
| TAG | 1.37 ± 0.93 |
| TC | 4.57 ± 1.06 |
| HDL | 1.37 ± 0.36 |
| LDL | 2.53 ± 0.76 |
| VLDL | 0.67 ± 0.49 |
| Thyroid function | |
| TSH | 2.47 ± 10.29 |
| TR-Ab | 12.29 ± 13.64 |
| TgAb | 120.93 ± 266.96 |

| Variables | Total ($n$ = 512) |
| --- | --- |
| TPOAb | 276.20 ± 568.75 |
| Tg | 364.85 ± 1,079.70 |
| Ultrasound characteristics | |
| Nodules in thyroid | |
| Yes | 470 (91.80) |
| No | 42 (8.20) |
| Aspect ratio imbalance | |
| Yes | 140 (27.34) |
| No | 372 (72.66) |
| Hypoechogenicity | |
| Yes | 247 (48.24) |
| No | 265 (51.76) |
| Irregular borders | |
| Yes | 221 (43.16) |
| No | 291 (56.84) |
| Irregular shape | |
| Yes | 95 (18.55) |
| No | 417 (81.45) |
| Microcalcifications | |
| Yes | 41 (8.01) |
| No | 471 (91.99) |

**Note:**
Data are expressed as the patient's number (%) or the mean ± standard deviation. A statistically significant difference was defined as $P < 0.05$. Abbreviations: BMI, body mass index; dx, diagnosis; GD, Graves' disease; HDL, high-density lipoprotein; KI, potassium iodide; LDL, low-density lipoprotein; MMI, Methimazole; PTU, propylthiouracil; TAG, triglycerides; TC, total cholesterol; Tg, Thyroglobulin; TgAb, antithyroglobulin antibodies; TPOAb, thyroid peroxidase antibodies; TR-Ab, TSH receptor antibody; TSH, thyroid-stimulating hormone; TT, total thyroidectomy; VLDL, very low-density lipoprotein.

(58.4%). The average age was 45.2, most (83.6%) of patients were women. The mean disease duration was 55.3 ± 79.1 months. Most patients (90.6%) underwent total or near-total thyroidectomy. The main cause of patients undergoing surgery was preoperative diagnosis of cancer or thyroid nodule (47.1%). The mean BMI was 23.01 ± 3.39. The average level of TAG, TC, HDL, LDL, and VLDL were 1.37 ± 0.93, 4.57 ± 1.06, 1.37 ± 0.36, 2.53 ± 0.76, and 0.67 ± 0.49 mmol/L, respectively. Most patients were found to have thyroid nodules during ultrasound evaluation (91.8%).

## Clinicopathological characteristics of GD patients with thyroid carcinoma

Table 2 presents the clinicopathological features of 299 individuals diagnosed with thyroid cancer. Among them, 296 patients (99.0%) were diagnosed with papillary thyroid carcinoma (PTC), whereas three patients (1.0%) were diagnosed with follicular thyroid carcinoma (FTC). Prior to surgery, 273 patients (91.3%) were diagnosed with thyroid cancer or nodules, while thyroid cancer was incidentally identified in 26 patients (8.7%).

**Table 2 Clinicopathological characteristics of thyroid cancer in Graves' disease.**

| Variables | Total ($n = 299$) |
|---|---|
| Incidentally discovered cancer | 26 (8.69) |
| Fine needle aspiration biopsy | 213 (71.2) |
| Cancer type | |
| PTC | 296 (99.02) |
| FTC | 3 (1.00) |
| Extent of surgery | |
| Lobectomy | 26 (8.70) |
| TT/near TT | 273 (91.30) |
| LN dissection | |
| No | 21 (7.02) |
| Central LN | 275 (91.98) |
| Central and lateral LN | 3 (1.00) |
| BRAF$^{V600E}$ | |
| Yes | 93/299 (31.10) |
| No | 12/299 (4.01) |
| Tumor size | |
| <10 mm | 220 (73.58) |
| ≥10 mm | 79 (26.42) |
| Multifocality | |
| Yes | 113 (37.79) |
| No | 186 (62.21) |
| Bilaterality | |
| No | 222 (74.25) |
| Yes | 77 (25.75) |
| ETE | |
| No | 172 (57.53) |
| Minimal | 118 (39.46) |
| Gross | 9 (3.01) |
| LNM | |
| No | 181 (60.54) |
| Yes | 97 (32.44) |
| AJCC 8th stage | |
| I | 278 (92.98) |
| II | 20 (6.69) |
| III | 1 (0.33) |

Note:
Abbreviations: AJCC, American Joint Committee on Cancer; BRAF, B-Raf proto-oncogene; ETE, extrathyroidal extension; FTC, follicular thyroid carcinoma; LN, lymph node; LNM, lymph node metastasis; PTC, papillary thyroid cancer.

Regarding surgical procedures, 91.3% of patients underwent total or near-total thyroidectomy, and 8.7% underwent lobectomy. Fine needle aspiration biopsy of the thyroid was conducted on 213 patients (71.2%) preoperatively. The tumor size of 73.6% of the patients was less than 10 mm. Multifocal tumors were observed in 113 patients

(37.8%), and bilaterality was noted in 77 patients (25.6%). A total of 127 patients (42.5%) exhibited extrathyroidal invasion, with 118 patients (39.5%) having minimal invasion and nine patients (3.0%) having gross invasion. Additionally, 275 patients (91.9%) underwent central lymph node dissection, and three patients (1.0%) underwent both central and lateral lymph node dissection. Lymph node metastasis was detected in 98 patients (32.8%). In terms of staging, 278 patients (93.0%) had stage I disease, 20 patients (7.4%) had stage II disease, and one patient (0.3%) had stage III disease, with a slight adjustment in percentage rounding for clarity.

## Association of serum lipid levels and thyroid carcinoma in patients with Graves' disease

To assess the connection between serum lipid concentrations and thyroid cancer among patients with Graves' disease, serum lipid markers were categorized based on optimal cutoff points derived from ROC curves (see Fig. S1). Table 3 illustrates that the proportion of overweight individuals was markedly greater in the GD/DTC+ cohort compared to the GD/DTC- cohort ($P < 0.001$). Furthermore, both TAG and VLDL levels were significantly elevated in the GD/DTC+ group *vs.* the GD/DTC- group ($P < 0.001$). Conversely, HDL levels were notably higher in the GD/DTC- group ($P < 0.001$). Additionally, various ultrasonic features exhibited a significant correlation with DTC in GD patients, including nodules in thyroid ($P < 0.001$), aspect ratio imbalance ($P < 0.001$), hypoechogenicity ($P < 0.001$), irregular borders ($P < 0.001$), microcalcifications ($P < 0.001$) and irregular shape ($P < 0.001$).

## Multivariate logistic regression analyses in GD patients combined with thyroid cancer

For the risk factors screened in Table 3, we selected those with *P*-values less than 0.01 for further univariate and multivariate regression analysis (Table 4). Univariate analysis revealed that, in patients with Graves' disease, levels of TAG, HDL and VLDL were significantly associated with thyroid cancer ($P < 0.001$). The lower HDL levels and high TAG and VLDL levels presented high risks of thyroid cancer. In addition, overweight, nodules in thyroid, aspect ratio imbalance, hypoechogenicity, irregular borders, microcalcifications and irregular shape were associated with higher risk of thyroid cancer in GD patients ($P < 0.001$).

Next, we incorporated the risk factors with a *P*-value less than 0.01 from the univariate regression analysis into the multivariate regression analysis. The multivariate analyses showed that higher levels of TAG (OR 2.55, 95% CI [1.39–4.71], $P = 0.003$) and lower levels of HDL (OR 0.41, 95% CI [0.24–0.70], $P = 0.001$) were independent risk predictors of thyroid cancer. Overweight (OR 2.51, 95% CI [1.34–4.71], $P = 0.004$), nodules in thyroid (OR 15.57, 95% CI [2.05–118.29], $P = 0.008$), aspect ratio imbalance (OR 16.65, 95% CI [6.21–44.62], $P < 0.001$), hypoechogenicity (OR 1.78, 95% CI [1.05–3.00], $P = 0.031$), irregular borders (OR 5.11, 95% CI [2.85–9.16], $P < 0.001$) and microcalcifications (OR 6.10, 95% CI [1.57–23.74], $P = 0.009$) also indicated high risk of thyroid cancer in patients with Graves' disease.

**Table 3 Cut-off of serum lipids in GD patients with or without thyroid cancer.**

| Variables | Total (*n* = 512) | GD without cancer (*n* = 213) | GD with cancer (*n* = 299) | *P* |
|---|---|---|---|---|
| Overweight | | | | <0.001 |
| No | 387 (75.59) | 182 (85.45) | 205 (68.56) | |
| Yes | 125 (24.41) | 31 (14.55) | 94 (31.44) | |
| TAG | | | | <0.001 |
| Low | 266 (51.95) | 151 (70.89) | 115 (38.46) | |
| High | 246 (48.05) | 62 (29.11) | 184 (61.54) | |
| TC | | | | 0.131 |
| Low | 118 (23.05) | 42 (19.72) | 76 (25.42) | |
| High | 394 (76.95) | 171 (80.28) | 223 (74.58) | |
| HDL | | | | <0.001 |
| Low | 247 (48.24) | 72 (33.80) | 175 (58.53) | |
| High | 265 (51.76) | 141 (66.20) | 124 (41.47) | |
| LDL | | | | 0.027 |
| Low | 211 (41.21) | 100 (46.95) | 111 (37.12) | |
| High | 301 (58.79) | 113 (53.05) | 188 (62.88) | |
| VLDL | | | | <0.001 |
| Low | 184 (35.94) | 98 (46.01) | 86 (28.76) | |
| High | 328 (64.06) | 115 (53.99) | 213 (71.24) | |
| Nodules in thyroid | | | | |
| Yes | 470 (91.80) | 172 (80.75) | 298 (99.67) | <0.001 |
| No | 42 (8.20) | 41 (19.25) | 1 (0.33) | |
| Aspect ratio imbalance | | | | |
| Yes | 140 (27.34) | 5 (2.35) | 135 (45.15) | <0.001 |
| No | 372 (72.66) | 208 (97.65) | 164 (54.85) | |
| Hypoechogenicity | | | | |
| Yes | 247 (48.24) | 55 (25.82) | 192 (64.21) | <0.001 |
| No | 265 (51.76) | 158 (74.18) | 107 (35.79) | |
| Irregular borders | | | | |
| Yes | 221 (43.16) | 26 (12.21) | 195 (65.22) | <0.001 |
| No | 291 (56.84) | 187 (87.79) | 104 (34.78) | |
| Irregular shape | | | | |
| Yes | 95 (18.55) | 21 (9.86) | 74 (24.75) | <0.001 |
| No | 417 (81.45) | 192 (90.14) | 225 (75.25) | |
| Microcalcifications | | | | |
| Yes | 41 (8.01) | 3 (1.41) | 38 (12.71) | <0.001 |
| No | 471 (91.99) | 210 (98.59) | 261 (87.29) | |

**Note:**
Abbreviations: HDL, high-density lipoprotein; LDL, low-density lipoprotein; TAG, triglycerides; TC, total cholesterol; VLDL, very low-density lipoprotein.
**Table 4 Univariate and multivariate analyses of malignancy risk factors.**

| Variables | Univariable cox | | Multivariable cox | |
|---|---|---|---|---|
| | OR (95% CI) | *P* | OR (95% CI) | *P* |
| Overweight | 2.66 [1.69–4.17] | <0.001 | 2.51 [1.34–4.71] | 0.004 |
| TAG | 3.86 [2.65–5.61] | <0.001 | 2.55 [1.39–4.71] | 0.003 |
| HDL | 0.37 [0.26–0.53] | <0.001 | 0.41 [0.24–0.70] | 0.001 |
| VLDL | 2.07 [1.44–2.98] | <0.001 | 1.00 [0.54–1.87] | 0.995 |
| Nodules in thyroid | 72.47 [9.88–531.44] | <0.001 | 15.57 [2.05–118.29] | 0.008 |
| Aspect ratio imbalance | 34.83 [13.95–86.99] | <0.001 | 16.65 [6.21–44.62] | <0.001 |
| Irregular borders | 13.70 [8.53–21.99] | <0.001 | 5.11 [2.85–9.16] | <0.001 |
| Irregular shape | 3.03 [1.80–5.10] | <0.001 | 1.54 [0.75–3.17] | 0.237 |
| Hypoechogenicity | 5.24 [3.56–7.71] | <0.001 | 1.78 [1.05–3.00] | 0.031 |
| Microcalcifications | 9.96 [3.03–32.72] | <0.001 | 6.10 [1.57–23.74] | 0.009 |

**Note:**
Abbreviations: HDL, high-density lipoprotein; TAG, triglycerides; VLDL, very low-density lipoprotein.

## Construction of nomogram for prediction of thyroid cancer in patients with Graves' disease

To develop a clinically practical predictive model, we stratified 512 patients into a training set ($n = 358$) and a validation set ($n = 154$), maintaining a 7:3 ratio. The baseline demographics of patients in both cohorts are presented in Table S1. Utilizing multivariate Cox proportional hazards regression analysis, we constructed a nomogram to assess the risk of thyroid cancer in individuals with Graves' disease (Fig. 2A). This nomogram highlighted aspect ratio >1 as the most influential predictor of thyroid cancer risk in Graves' disease patients. Additionally, TAG levels, HDL levels, overweight status, thyroid nodules, hypoechogenicity, irregular borders, and microcalcifications emerged as crucial factors. Calibration plots for both cohorts demonstrated good accuracy of the nomogram in predicting thyroid cancer among Graves' disease patients (Fig. 2B). To further validate the model's discriminatory power, ROC curves were generated. The area under the curve (AUC) for the predictive model was 0.91 (95% CI [0.88–0.94]) in the training cohort and 0.91 (95% CI [0.87–0.96]) in the validation cohort, respectively (Figs. 2C, 2D). In the training cohort, the sensitivity/specificity of the model was 0.826/0.857. In the validation cohort, the sensitivity/specificity of the model was 0.844/0.831. In addition, to further validate the robustness of the model, we performed a 5-fold cross-validation (Fig. S2). The results showed that our model also exhibited good predictive performance under 5-fold cross-validation (average AUC: 0.906).

## The association between serum lipids and clinicopathological characteristics of GD patients with PTC

To investigate the link between serum lipid concentrations and clinicopathological features in patients with papillary thyroid cancer (PTC) coexisting with Graves' disease (GD), we stratified the 299 patients with thyroid cancer into high and low groups according to the median values of BMI, TAG, and HDL. Notably, elevated TAG levels exhibited a

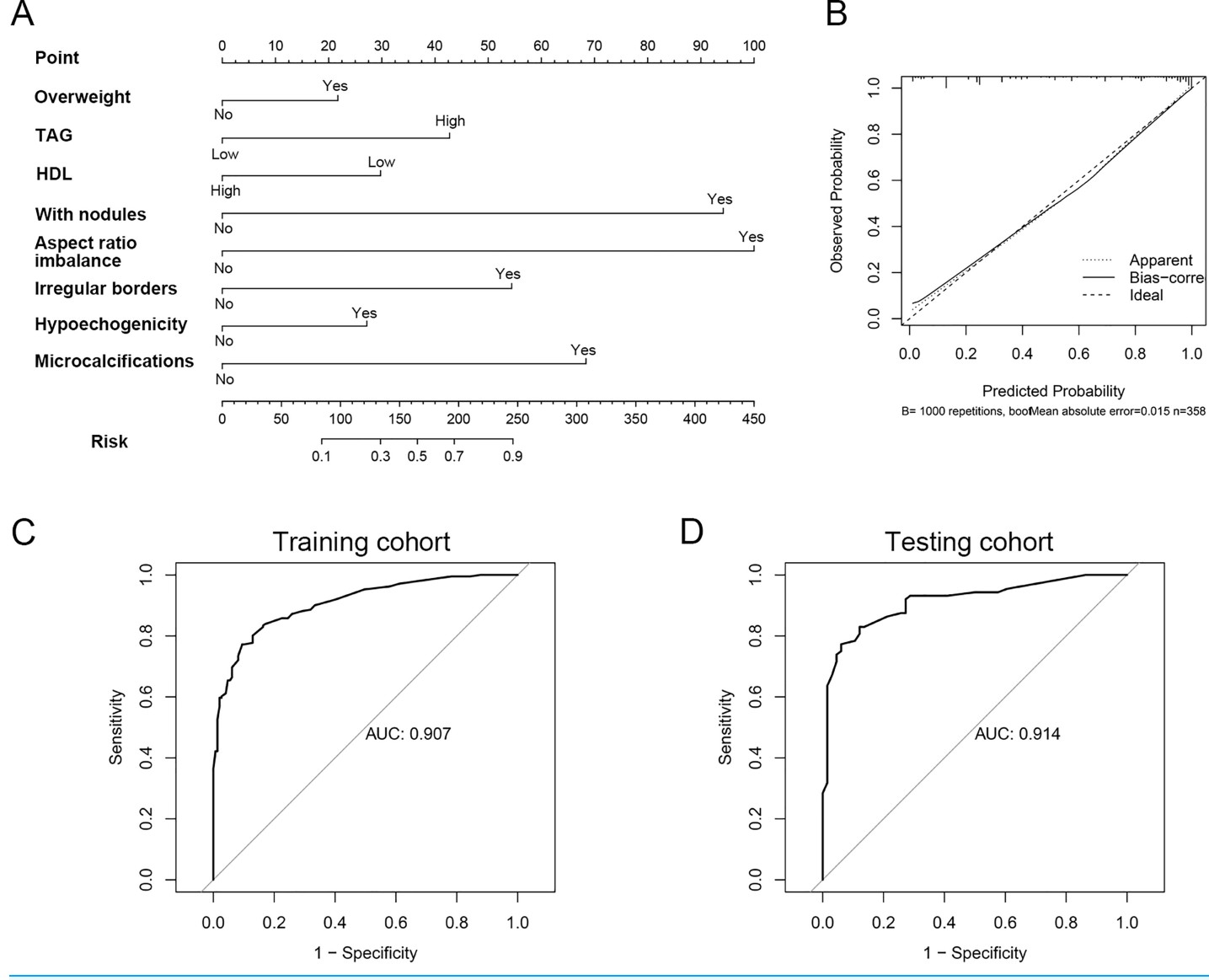

**Figure 2 The model constructed for diagnosis of thyroid cancer in patient with GD.** (A) Nomogram constructed for diagnosis of thyroid cancer. (B) Calibration curve for nomogram. (C) The ROC curve of the efficacy of nomogram model in the training cohort. (D) The ROC curve of the efficacy of nomogram model in the validation cohort.

significant correlation with lymph node metastasis (LNM) ($P = 0.033$) and the American Joint Committee on Cancer (AJCC) 8th stage ($P = 0.015$). However, no statistically significant differences were observed in tumor size, multifocality, bilaterality, or extrathyroidal extension (ETE) between the TAG groups (Table 5). Conversely, no discernible differences in clinicopathological characteristics were noted between the high and low BMI or HDL groups (Tables S2, S3). Subsequently, a comparative analysis was conducted to investigate the potential associations between different thyroid stimulating hormone (TSH)/Trab levels and lipid profiles. Statistical analyses revealed that patients with abnormal TSH levels (either hypo- or hyperthyrotropic) exhibited significantly

**Table 5 Clinical characteristics of the high TAG and low TAG groups.**

| Variables | TAG | | |
|---|---|---|---|
| | Low | High | *P* |
| Age (>55) | | | 0.104 |
| <55 | 118 (78.67) | 105 (70.47) | |
| ≥55 | 32 (21.33) | 44 (29.53) | |
| BRAF[V600E] | | | 0.661 |
| No | 5 (10.00) | 7 (12.73) | |
| Yes | 45 (90.00) | 48 (87.27) | |
| Tumor size | | | 0.868 |
| <10 mm | 111 (74.00) | 109 (73.15) | |
| ≥10 mm | 39 (26.00) | 40 (26.85) | |
| Multifocality | | | 0.869 |
| No | 94 (62.67) | 92 (61.74) | |
| Yes | 56 (37.33) | 57 (38.26) | |
| Bilaterality | | | 0.337 |
| No | 115 (76.67) | 107 (71.81) | |
| Yes | 35 (23.33) | 42 (28.19) | |
| ETE | | | 0.351 |
| No | 90 (60.00) | 82 (55.03) | |
| Minimal | 54 (36.00) | 64 (42.95) | |
| Gross | 6 (4.00) | 3 (2.01) | |
| LNM | | | 0.033 |
| No | 97 (71.32) | 84 (59.15) | |
| Yes | 39 (28.68) | 58 (40.85) | |
| AJCC 8th stage | | | 0.015 |
| I | 145 (96.67) | 133 (89.26) | |
| II | 5 (3.33) | 15 (10.07) | |
| III | 0 (0.00) | 1 (0.67) | |

Note:
Abbreviations: AJCC, American Joint Committee on Cancer; BRAF, B-Raf proto-oncogene; ETE, extrathyroidal extension; LNM, lymph node metastasis.

elevated TAG levels compared to those within the normal TSH range (Table S4). However, no significant differences were observed in other lipid parameters. Regarding Trab levels, no significant variations in any lipid markers were detected between the low- and high-Trab groups. Furthermore, we performed a subgroup analysis comparing TSH and Trab levels between patients with and without thyroid cancer (Table S5). The results of this analysis demonstrated that no discernible differences in TSH or Trab level were noted between GD patients with or without thyroid cancer (Table S6).

## DISCUSSION

The correlation between serum lipids and thyroid cancer in individuals with Graves' disease remains unclear. In our study, we initially uncovered that TAG and HDL cholesterol levels independently indicated an elevated risk of thyroid cancer in patients

with GD. By integrating TAG and HDL with various clinicopathological factors, we subsequently developed and validated an innovative nomogram for predicting the risk of thyroid cancer in this patient population.

Reports indicated that thyroid cancer occurs in 1.7% to 11.5% of GD patients (*Soares et al., 2023*; *Papanastasiou et al., 2019*; *Yoon et al., 2021*). Given that our study focused on GD patients meeting surgical criteria, the cohort demonstrated a thyroid cancer prevalence higher than that of the general GD population, yet comparable to the general population's incidence. All thyroid cancer subtypes can manifest in GD patients. Although GD primarily does not necessitate surgical management, surgical intervention becomes an option when specific criteria are met. According to the 2016 American Thyroid Association (ATA) guidelines for hyperthyroidism, near-total or total thyroidectomy is advised for surgical treatment of GD. However, in our study, 48 patients underwent lobectomy due to preoperatively detected nodules and based on the joint decision of the patients and the multidisciplinary team. Notably, no recurrence of GD was observed in the 48 patients who underwent lobectomy.

Numerous studies have confirmed a correlation between abnormal serum lipids and the occurrence and development of cancer. TAG, a key component of lipid droplets, is essential for the storage and transport of fatty acids, as well as for facilitating cellular energy utilization. Previous studies have established a link between serum TAG concentrations and the onset of PTC (*Papanastasiou et al., 2019*). In PTC, serum TAG is a risk factor for tumors larger than 5 mm, multiple lesions, accompanied by ETE, and distant metastasis (*Nguyen, Kim & Kim, 2022*). *Zeng et al. (2024)* showed that abnormal accumulation of TAG was observed in lymphatic metastases, but not in normal thyroid tissue, primary PTC, or healthy lymph nodes (*Li et al., 2021*). HDL, a key lipoprotein involved in blood cholesterol transport, is composed primarily of cholesterol. A decreased HDL level has emerged as a risk factor for thyroid cancers. *Park et al. (2024)* found that a lower HDL concentration was associated with a heightened risk of PTC, with a hazard ratio of 1.17 (95% CI [1.15–1.19]), suggesting a notable correlation with PTC incidence (*Zeng et al., 2024*; *Park et al., 2020*). The serum HDL level is significantly negatively correlated with the occurrence and development of thyroid cancer (*Xu et al., 2021*). Additionally, research has indicated that thyroid cancer tends to be more aggressive in patients who are obese or overweight, irrespective of the coexistence of GD. *Kitahara et al. (2011)* reported a positive correlation between BMI and thyroid cancer risk in both genders. Furthermore, *Park et al. (2024)* found that overweight can serve as a predictive biomarker for thyroid cancer in individuals with GD, with an odds ratio of 3.108 (95% CI [1.196–8.131]; $P = 0.021$). Note the slight adjustment in the confidence interval endpoint for accuracy. GD is a hypermetabolic disorder that typically leads to a decrease in blood lipids (including TAG, TC, and LDL) and weight loss. During the transition from hyperthyroidism to euthyroidism in Graves' disease, an increase in weight and blood lipids is usually observed (*Chng et al., 2016*; *Altinova et al., 2006*). Therefore, monitoring blood lipids and weight simultaneously can serve as one of the indicators for assessing the condition of GD and observing therapeutic efficacy. However, studies have shown that although obese patients experience greater weight loss when diagnosed with GD, they remain morbidly obese and

have higher levels of thyroid hormones compared to non-obese patients (*Hoogwerf & Nuttall, 1984*).

The mechanisms by which blood lipid levels and obesity influence the development of thyroid cancer in patients with GD remain unclear. It is currently known that the role of triglycerides in tumors is related to the associated hydrolase, monoacylglycerol lipase (MAGL). Proteomic studies of hydrolases in various cancer cell lines have consistently shown upregulated expression of MAGL in malignant cells of melanoma, ovarian cancer, and breast cancer compared to non-malignant cells. Exogenous overexpression of this gene can promote tumor infiltration, while inhibiting its expression can suppress tumor growth and metastasis (*Hu et al., 2014*). In addition, insulin resistance plays a pivotal role in the development of thyroid cancer. Metabolic syndrome (MetS) and its components are associated with an increased risk and aggressiveness of thyroid cancer. Insulin resistance, aided by adipokines, angiotensin II, and estrogen, exerts a central role in promoting the progression of thyroid cancer. Triglycerides, as a key component of the TyG index (Triglyceride-Glucose index) reflecting insulin resistance, can activate the Akt signaling pathway through G-protein-coupled receptors, indirectly facilitating the growth and development of thyroid cancer cells. For instance, in the state of insulin resistance, hyperinsulinemia occurs, activating the PI3K/Akt/mTOR/S6R signaling pathway, which facilitates glucose uptake and utilization by tumor cells, providing energy for cancer cell proliferation (*Li et al., 2023*). Controversy also exists about the association between HDL levels. From a molecular perspective, various hypotheses have been proposed to explain the protective role of HDL against cancer. Firstly, HDL contains an antioxidant enzyme called paraoxonase-1, which is believed to inhibit tumorigenesis by reducing oxidative stress in peripheral tissues (*Vekic et al., 2007*). Second, reduced HDL level might be a secondary phenomenon driven by cancer cell metabolism. Malignant cells can induce liposynthesis and accumulation of intracellular cholesteryl esters for new membrane biogenesis. The scavenger receptor class B type (SR-BI), an HDL-C receptor, is highly expressed on tumor cell surfaces. This receptor facilitates the uptake of cholesteryl esters from HDL-C into the cytoplasm, leading to a reduction in plasma HDL-C (*Tosi & Tugnoli, 2005*). Furthermore, high SR-BI expression was related to aggressive tumors and poor clinical outcomes (*Yuan et al., 2016*).

Ultrasound is the most important diagnostic tool for evaluating thyroid cancer in patients with GD. In this study, several ultrasound indicators can predict the risk of thyroid cancer in GD patients, including the presence of ultrasound-visible thyroid nodules, loss of anterior-posterior to transverse diameter ratio, blurred margins, irregular margins, and microcalcifications. It is noteworthy that the majority of GD patients have thyroid nodules, yet approximately 9% of thyroid cancer patients do not have suspicious malignant thyroid nodules detected by preoperative ultrasound. In this study, a predictive model incorporating blood lipids, BMI, and ultrasound characteristics further increased the detection rate of thyroid cancer in GD patients, providing additional evidence for determining whether further surgical treatment is needed for GD patients.

This study, due to its retrospective nature, has several inherent limitations. Firstly, the inclusion of only a single-center cohort of surgically treated GD patients may not provide an accurate representation of the true thyroid cancer incidence among all GD patients. Secondly, there is selection bias in this study due to surgical cohort. In addition, the baseline blood lipid measurements were taken just once prior to surgery, potentially limiting the accuracy of assessing the relationship between blood lipids and thyroid cancer in GD patients. And the heterogeneity in surgical indications among our patients may introduce confounding factors that could affect the generalizability of our findings. Moreover, the duration of follow-up for thyroid cancer was insufficient to comprehensively evaluate long-term clinical outcomes. Therefore, large-scale, prospective studies are imperative to rigorously investigate the link between thyroid cancer and GD.

## CONCLUSIONS

This study initially demonstrated that preoperative serum lipids and body mass index were independent indicators for thyroid cancer in patients with GD. The predictive model constructed by BMI, serum lipid profiles (TAG and HDL) along with ultrasound characteristics (nodules in thyroid, aspect ratio imbalance, hypoechogenicity, irregular borders, and microcalcifications) facilitate clinicians to screen GD patients with a higher risk of developing thyroid cancer.

### Funding
The authors received no funding for this work.

### Competing Interests
The authors declare that they have no competing interests.

### Author Contributions
- Xingxing Gao conceived and designed the experiments, performed the experiments, analyzed the data, prepared figures and/or tables, authored or reviewed drafts of the article, and approved the final draft.
- Mi Liu performed the experiments, analyzed the data, prepared figures and/or tables, and approved the final draft.
- Yijun Wu conceived and designed the experiments, authored or reviewed drafts of the article, and approved the final draft.

### Human Ethics
The following information was supplied relating to ethical approvals (*i.e.*, approving body and any reference numbers):

Studies involving human participants were reviewed and approved by the Clinical Research Ethics Committee of the First Affiliated Hospital College of Medicine, Zhejiang University (2024-1193).

## Data Availability

Raw data is available in the Supplemental Files.

## Supplemental Information

Supplemental information for this article can be found online at http://dx.doi.org/10.7717/peerj.19915#supplemental-information.

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
