# Peer review of "Preoperative serum lipids as novel predictors for concomitant thyroid carcinoma in Graves’ disease"

_PeerJ, doi:10.7717/peerj.19915_

## Round 0.1 · original submission · Major Revisions

Reviewer 1 ·

Basic reporting

The manuscript investigates the association between preoperative serum lipids, BMI, and thyroid carcinoma in Graves’ disease (GD) patients, proposing a predictive nomogram. The study addresses a clinically relevant gap, as the link between metabolic factors and thyroid cancer in GD remains underexplored. The retrospective design, large sample size, and rigorous statistical methods are strengths. However, several issues require clarification to enhance the manuscript’s impact and validity.

Experimental design

1. Ethics Approval: Stated but lacks IRB approval number or waiver specifics.
2.Inclusion/Exclusion Criteria: Well-defined, but the exclusion of anaplastic/medullary cancers should be justified, as their rarity in GD might not significantly bias results.
3 . Retrospective Limitations: Acknowledge potential biases (e.g., selection bias due to surgical cohort).

Validity of the findings

1. Statistical Robustness:
The AUC (0.91) is impressive but raises questions about overfitting. Provide sensitivity/specificity metrics and consider cross-validation.
Clarify why VLDL, significant in univariate analysis, was non-significant in multivariate analysis.
2 .Confounders:
Address potential confounders (e.g., medication effects on lipids, duration of GD).
Thyroid hormone levels (TSH, TR-Ab) are reported but not analyzed for correlations with lipids/cancer.

·

Basic reporting

This manuscript generally meets our journal's standards in terms of language, literature references, and article structure. But authors should pay attention to the standardized use of abbreviations. The full terms should be provided when they first appear, and then the abbreviations should be used consistently throughout the text instead of the full terms, such as DTC, etc.
Minor issues exist (e.g., Table 1, line 7, “thyroidetomy”). The research is based on solid background knowledge, but the reference should be more detailed (e.g., please provide the reference information of line 83, “the annual incidence of thyroid carcinoma among GD patients has been escalating”).
Figures and tables are well-structured, while the raw data are available upon request, I suggest providing a supplementary file including necessary data repository or standardized metadata.

Experimental design

The experiment was well designed, but the followings should be noticed.
1. Table 3 includes information on 512 patients, but I noticed that the results for LDL levels are “Low 197 (41.21) and High 281 (58.79)” which includes 478 participants. Please explain in the methodology or other appropriate sections why there is data missing and how you have dealt with the missing data.
2. The study included 512 patients, a sufficiently large sample size. However, the fact that all patients were recruited from a single center might restrict the generalizability of the results. It is recommended to acknowledge this limitation in the discussion section and to incorporate more data in future research.
3. Since the study is based on GD patients meeting surgical criteria, the specific situation each patient faced, and what necessitate him/her a surgery may be an influencing factor.

Validity of the findings

1. The results represent high validation, since the AUC for the predictive model was 0.91 in training/validation cohort.
2. The conclusions correspond to statistical results. However, in the discussion, authors have listed several observational research in accord with the present results. More mechanistic study focused on casual relationship should be expanded.

Reviewer 3 ·

Basic reporting

see additional comments

Experimental design

see additional comments

Validity of the findings

see additional comments

Additional comments

The manuscript investigates the relationship between preoperative serum lipid levels and the presence of concomitant thyroid carcinoma in patients with Graves' disease. The study is well-structured and methodologically sound. Below are my comments:
1. The manuscript would benefit from moderate language refinement. The authors are encouraged to use a language editing service or a large language model to improve clarity, grammar, and flow.
2. Were thyroid function markers such as FT3 and FT4 collected during the study? If so, please specify this in the Methods section and clarify whether these were considered in the analysis.
3. discussion: In the limitations section, the authors should acknowledge that the validation cohort is internal, and no external validation was conducted. This limitation affects the generalizability of the findings and should be explicitly discussed.
4. Were there any missing data in the clinical or laboratory parameters? If so, please describe how missing data were handled in the analysis.

---

## Round 0.2 · accepted · Accept

Dear authors,

Thank you very much for addressing correctly all the issues raised by the reviewers. This manuscript can now be recommended for publication.
Best regards,

Fernando

Reviewer 1 ·

Basic reporting

-

Experimental design

-

Validity of the findings

-

·

Basic reporting

-

Experimental design

-

Validity of the findings

-

Additional comments

The revised one has dealt with my concerns.

Reviewer 3 ·

Basic reporting

-

Experimental design

-

Validity of the findings

-

Additional comments

no further comments